# Cancer Metastasis Detection With Neural Conditional Random Field

**Yi Li**
Baidu Silicon Valley Artificial Intelligence Lab
1195 Bordeaux Dr. Sunnyvale, CA 94089
liyi17@baidu.com

**Wei Ping**
Baidu Silicon Valley Artificial Intelligence Lab
1195 Bordeaux Dr. Sunnyvale, CA 94089
pingwei01@baidu.com

## Abstract

Breast cancer diagnosis often requires accurate detection of metastasis in lymph nodes through Whole-slide Images (WSIs). Recent advances in deep convolutional neural networks (CNNs) have shown significant successes in medical image analysis and particularly in computational histopathology. Because of the outrageous large size of WSIs, most of the methods divide one slide into lots of small image patches and perform classification on each patch independently. However, neighboring patches often share spatial correlations, and ignoring these spatial correlations may result in inconsistent predictions. In this paper, we propose a neural conditional random field (NCRF) deep learning framework to detect cancer metastasis in WSIs. NCRF considers the spatial correlations between neighboring patches through a fully connected CRF which is directly incorporated on top of a CNN feature extractor. The whole deep network can be trained end-to-end with standard back-propagation algorithm with minor computational overhead from the CRF component. The CNN feature extractor can also benefit from considering spatial correlations via the CRF component. Compared to the baseline method without considering spatial correlations, we show that the proposed NCRF framework obtains probability maps of patch predictions with better visual quality. We also demonstrate that our method outperforms the baseline in cancer metastasis detection on the Camelyon16 dataset and achieves an average FROC score of 0.8096 on the test set. NCRF is open sourced at https://github.com/baidu-research/NCRF.

## 1 Introduction

Breast cancer is one of the leading causes of death among women in the United States [25]. Early cancer diagnosis and treatment play a crucial role in improving patients' survival rate [23]. One of the most important early diagnosis is to detect metastasis in lymph nodes through microscopic examination of hematoxylin and eosin (H&E) stained histopathology slides. In recent years, pathologists have been using Whole-slide Images (WSIs) to distinguish between normal and tumor cells and localize malignant lesions [3]. However, manually detecting tumor cells within extremely large WSIs (e.g., $100{,}000 \times 200{,}000$ pixels) can be tedious and time-consuming. Significant discordance on detection results among different pathologists has also been reported [8], where the overall concordance rate of participating pathologists was $75.3\%$. Therefore, various computer-aided diagnosis (CAD) systems have been developed to assist pathologists to detect cancer metastasis in WSIs [6, 2].

1st Conference on Medical Imaging with Deep Learning (MIDL 2018), Amsterdam, The Netherlands.

In recent years, deep convolutional neural networks (CNNs) have shown significant improvements on a wide range of computer vision tasks on natural images, e.g. image classification [17, 24, 12], object detection [11, 10], and semantic segmentation [19]. Similarly, a few promising studies have also applied deep CNNs to analyse medical images and particularly WSIs [27, 9, 13, 18, 26, 31, 30], among which [27] won the Camelyon16 challenge [1] for metastasis detection. Because of the extremely large size of WSIs, most of the studies first extracted small patches (e.g. $256 \times 256$ pixels) from WSIs, and trained a deep CNN to classify these small patches into normal or tumor regions. A probability map of the original WSI being tumor or normal at patch level was later obtained and metastasis detection was performed based on this probability map. However, the small patches and their neighbors often share spatial correlations. Because the patches were extracted and trained independently, the spatial correlations were not modeled explicitly. Therefore, during inference time, the predictions over neighboring patches may be inconsistent, and the patch level probability map may contain isolated outliers [15, 28].

To explicitly model the spatial correlations between neighboring patches, Kong et al. [15] recently proposed Spatio-Net that uses 2D Long Short-Term Memory (LSTM) layers to capture the spatial correlations based on patch features extracted from a CNN classifier. However, Spatio-Net uses a two-stage training approach, and therefore the CNN feature extractor is not aware of the spatial correlations [28]. In parallel to our work and very recently, we notice a similar work [28] that also uses features extracted from a CNN classifier to represent neighboring patches. Conditional random field (CRF) is then applied on these patch features to model spatial correlations and refine the predicted probability map during a post-processing stage. In addition to the same issue due to the two stage framework, there is a significant computational overhead during the CRF post-processing, and the authors have to select a limited number of features (e.g., 5 reported in [28]) from the original high dimensional patch representations to perform the CRF inference algorithm on CPU.

In this paper, we propose an alternative method for modeling spatial correlations between neighboring patches through neural conditional random field (NCRF). NCRF is a probabilistic graphical model that combines both neural networks and conditional random fields. It has been used for sequence labeling [4] and semantic image segmentation [7]. We implemented NCRF based on the idea of conditional random fields as recurrent neural networks [29] that directly incorporates a fully connected CRF on top of the CNN feature extractor. The marginal label distribution of each patch is obtained through the mean-field approximate inference algorithm. The whole deep network can be trained in an end-to-end manner with the standard back-propagation algorithm, avoiding the post-processing stage. Because the mean-field inference algorithm is also performed on GPU, the CRF component introduces minor computational overhead and allows very large feature dimensions, e.g. 512 from the ResNet architecture [12]. The CNN feature extractor also benefits from jointly training with the CRF component, since it is now aware of the spatial correlations between neighboring patches. Compared to the baseline method that does not consider patch spatial correlations, we show that, 1) NCRF improves the visual quality of the probability map, 2) NCRF improves the CNN feature extractor, and 3) NCRF improves the performance of cancer metastasis detection. On the test set of the Camelyon16 challenge, the best average free response receiver operating characteristic (FROC) score of NCRF is 0.8096, which outperforms the previous best average FROC score of 0.8074 reported in [27].

## 2 Method

In this section, we describe the details of the proposed neural conditional random field (NCRF) model. Figure 1 shows the overall architecture of NCRF. It has two major components: CNN and CRF. The CNN component acts as a feature extractor, that takes a grid of patches as input, and encodes each patch as a fixed-length vector representation (i.e. embedding). The CRF component takes the grid of embeddings as input and models their spatial correlations. The final output from the CRF component is the marginal distribution of each patch being normal or tumor given the grid of patch embeddings. We illustrate the details of each component in the next two sections.

### 2.1 Patch Embedding With CNN

To extract comprehensive feature representation of each patch, we employ two ResNet architectures [12] that have proven to be powerful in image classification task, ResNet-18 and ResNet-34. For each architecture, we use the activations after the average pooling layer as the embedding for

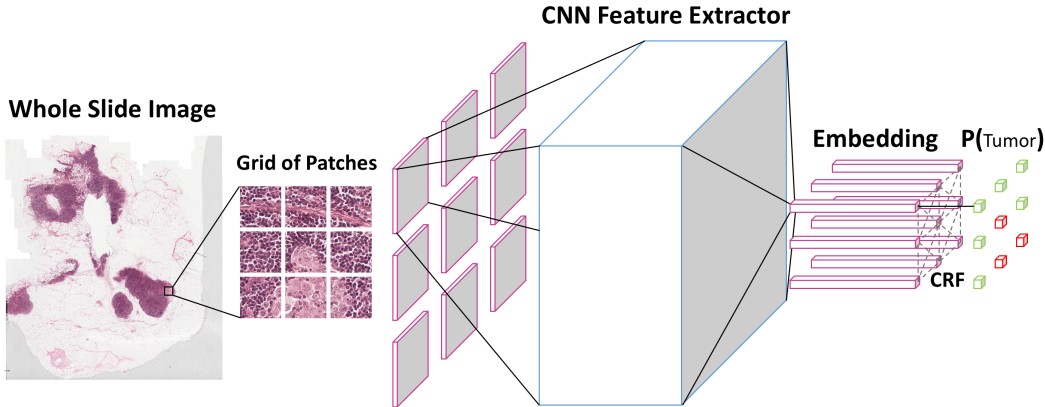

Figure 1: The architecture of NCRF model.

each patch. The embedding size is 512 for both ResNet-18 and ResNet-34, which is much larger than the embedding size of 5 reported in [28] after feature selection.

## 2.2 Spatial Modeling With CRF

In this section, we describe the methodology details of the CRF component. We denote a grid of patch embeddings obtained from CNN as $\boldsymbol{x} = \{x_i\}_{i=1}^N$, where $N$ is the number of patches within the grid, e.g. 25 for a grid of $5 \times 5$. Let $\mathbf{Y} = \{y_i\}_{i=1}^N$ be the random variables associated with each patch $i$, that represents the label of patch $i$ takes a value from $\{normal, \ tumor\}$. The conditional distribution $P(\mathbf{Y} \mid \boldsymbol{x})$ can be modeled as a CRF with a Gibbs distribution of

$$P(\mathbf{Y} = \boldsymbol{y} \mid \boldsymbol{x}) = \frac{1}{Z(\mathbf{x})} \exp(-E(\boldsymbol{y}, \boldsymbol{x})) \tag{1}$$

where $E(\boldsymbol{y}, \boldsymbol{x})$ is the energy function that measures the cost of $\mathbf{Y}$ taking a specific configuration $\boldsymbol{y}$ given $\boldsymbol{x}$, and $Z(\boldsymbol{x})$ is the partition function that insures $P(\mathbf{Y} = \boldsymbol{y} \mid \boldsymbol{x})$ is a valid probability distribution. In a fully-connected pairwise CRF [16], the energy function is given by:

$$E(\mathbf{y}, \boldsymbol{x}) = \sum_i \psi_u(y_i) + \sum_{i<j} \psi_p(y_i, y_j) \tag{2}$$

where $i, j$ ranges from 1 to $N$. $\psi_u(y_i)$ is the unary potential that measures the cost of patch $i$ taking the label $y_i$ given the patch embedding $x_i$, and $\psi_p(y_i, y_j)$ is the pairwise potential that measures the cost of jointly assigning patch $i, j$ with label $y_i, y_j$ given the patch embeddings $x_i, x_j$. Pairwise potential $\psi_p(y_i, y_j)$ models spatial correlations between neighboring patches, and would encourage low cost for assigning $y_i, y_j$ with the same label if $x_i, x_j$ are similar. We implement the unary potential $\psi_u(y_i)$ as the negative log-likelihood of patch $i$ taking label $y_i$, which is the negative logit for label $y_i$ before the softmax layer of the CNN classifier. We implement the pairwise potential as the weighted cosine distance between $x_i, x_j$:

$$\psi_p(y_i, y_j) = \mathbb{I}(y_i = y_j) \cdot w_{i,j} \left(1 - \frac{x_i \cdot x_j}{\|x_i\| \, \|x_j\|}\right) \tag{3}$$

where $\mathbb{I}(y_i = y_j)$ is the indicator function that checks the label compatibility between $y_i, y_j$, and $w_{i,j}$ is a trainable weight which controls the correlation strength between two patches $i, j$ within the grid. Typically, fully connected CRF also includes another distance term for pairwise potential that encodes the spatial distance between two patches $i, j$ [16]. However we did not observe clear improvements by including such distance term, and if we put a trainable coefficient before the term, the coefficient was pushed to zero during training. On the other hand, we observed the trainable weight $w_{i,j}$ correlated well with the relative distances between different patches within the grid after model converged, and we show this in the results section.

In order to train the CNN-CRF architecture end-to-end with the standard back-propagation algorithm, we need to obtain the marginal distribution of each patch label $y_i$, so that it can be used to compute the

cross-entropy loss with respect to the ground truth labels [29]. However, exact marginal inference is intractable, and we use mean-field approximate inference, where the original CRF distribution $P(\mathbf{Y})$[1] is approximated with a simpler distribution $Q(\mathbf{Y})$, that can be written as the product of marginal distributions of each individual patch $i$, $Q(\mathbf{Y}) = \prod_i^N Q_i(y_i)$. By minimizing the KL divergence between $Q(\mathbf{Y})$ and $P(\mathbf{Y})$, $\mathbb{KL}(Q(\mathbf{Y}) \| P(\mathbf{Y}))$, we derive the update step for each marginal distribution $Q_i(y_i)$ [20]:

$$\log Q_i(y_i) = \mathbb{E}_{-Q_i}\left[\log \tilde{P}(\mathbf{Y})\right] + \text{const} \tag{4}$$

where $\mathbb{E}_{-Q_i}[f(\mathbf{Y})]$ means taking the expectation of $f(\mathbf{Y})$ with respect to all the variables except $y_i$, and $\tilde{P}(\mathbf{Y}) = \exp(-E(\boldsymbol{y}, \boldsymbol{x}))$ is the unnormalized CRF distribution. The mean-field inference algorithm is summarized in Algorithm 1

---

**Algorithm 1** Mean-field inference algorithm

---

compute $\psi_u(y_i)$ for all $i$ and $\psi_p(y_i, y_j)$ for all $i, j$
$\log \tilde{P}(\mathbf{Y}) \leftarrow -\left[\sum_i \psi_u(y_i) + \sum_{i<j} \psi_p(y_i, y_j)\right]$
initialize $Q_i(y_i) \leftarrow \exp(-\psi_u(y_i))$ for all $i$
normalize $Q_i(y_i)$ for all $i$
**for** T iterations **do**
    $\log Q_i(y_i) \leftarrow \mathbb{E}_{-Q_i}\left[\log \tilde{P}(\mathbf{Y})\right]$ for all $i$
    normalize $Q_i(y_i)$ for all $i$
**end for**

---

Finally, after a fixed number of mean-field iterations, we use the approximate marginal distribution of each patch label $Q_i(y_i)$ to compute the cross-entropy loss and train the whole model with back-propagation algorithm.

## 3 Experiments

In this section, we present empirical evaluations on the proposed NCRF method. We demonstrate its advantages over the baseline method without CRF in three aspects: 1) NCRF obtains smoother probability maps with sharp boundaries than the baseline method, 2) NCRF achieves higher patch level classification accuracies from the CNN feature extractor than the baseline method, and 3) NCRF outperforms the baseline method in cancer metastasis detection.

### 3.1 Data Preparation

We conducted all the experiments based on the Camelyon16 dataset [1, 5]. This dataset includes 160 normal and 110 tumor WSIs for training, 81 normal and 49 tumor WSIs for testing. Slides were exhaustively annotated by pathologists in pixel level, with a few exceptions reported in [18]. We conducted all the experiments on $40\times$ magnification. We used the Otsu algorithm [21, 27] to exclude the background regions of each training slide. We used Normal_001 to Normal_140 and Tumor_001 to Tumor_100 for training, and the rest of training slides for validation. To generate patches, we first randomly picked one slide and then randomly sampled a coordinate from the slide as the center of the patch. We randomly sampled $200,000$ $768 \times 768$ patches from the tumor regions of the tumor slides as positive samples. We randomly sampled $200,000$ $768 \times 768$ patches from the non-tumor non-background regions of the tumor slides and the non-background regions of the normal slides as negative samples. Hard negative mining [27] was also applied to sample more patches from the tissue boundary regions.

### 3.2 Implementation Details

NCRF was implemented with PyTorch-0.3.1 [22] and trained with NVIDIA GeForce GTX 1080 Ti GPU. The mean-field inference algorithm for the CRF component was performed 10 iterations for

---

[1]We omit the dependency on $\boldsymbol{x}$ here for clarity.

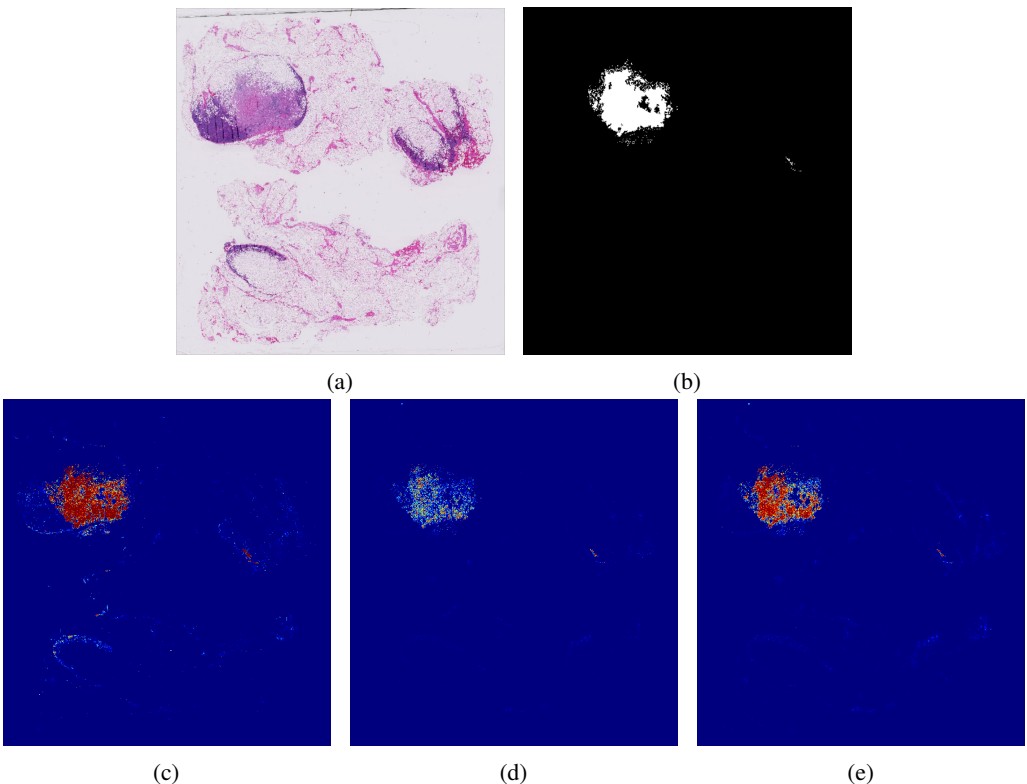

| (a) | (b) |

| (c) | (d) | (e) |

Figure 2: Predicted probability maps of Test_026 from ResNet-18 with random seed 0. (a) original WSI, (b) ground truth annotation, (c) baseline method, (d) baseline method with hard negative mining, (e) NCRF with hard negative mining

all the architectures. The CRF component introduces less than 0.1 seconds computational overhead per batch iteration, since the mean-field inference algorithm is also performed on GPU. During the training stage, a batch size of $20\ 768 \times 768$ patches were feed into the model. Each $768 \times 768$ patch was further split into a $3 \times 3$ grid of $256 \times 256$ patches and their corresponding labels were retrieved. The pixel values of patches were normalized by subtracting 128 and dividing 128. During training, color jitter was added using torchvision tranforms with parameters adopted from [18] :brightness with a maximum delta of 64/255, contrast with a maximum delta of 0.75, saturation with a maximum delta of 0.25, and hue with a maximum delta of 0.04. Patches were also randomly flipped and rotated with multiplies of 90°. We used stochastic gradient descent of learning rate 0.001 and a momentum of 0.9 to optimize all the architectures for 20 epochs. Each architecture was repetitively trained 5 times with different random seeds for parameters initialization. During inference time, probability maps were generated with a stride of 64 (level 6).

## 3.3 NCRF Obtains Smooth Probability Maps

Figure 2 shows the predicted probability maps of Test_026 from the baseline method, baseline method with hard negative mining, and NCRF with hard negative mining, all based on the ResNet-18 architecture with random seed 0. We can see the probability map from the baseline method that does not consider spatial correlations tends to contain isolated outlier predictions, which significantly increases the number of false positives. Hard negative mining significantly reduces the number of false positives for the baseline method, but the probability density among the ground truth tumor regions is also decreased, which decreases model sensitivity. Compared to the baseline method with hard negative mining, NCRF with hard negative mining not only achieves low false positives but also maintains high probability density among the ground truth tumor regions with sharp boundaries. In fact, NCRF detects two more tumor regions in Test_026 compared to the baseline method in this case.

## 3.4 NCRF Improves CNN Feature Extractor

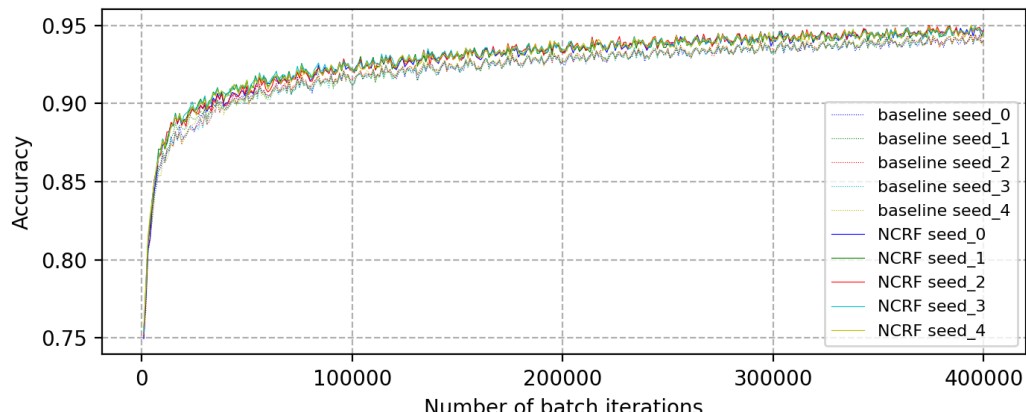

Figure 3: Patch classification accuracies of the baseline method and NCRF based on the ResNet-18 architecture throughout training with different random seeds for initialization, best view in color.

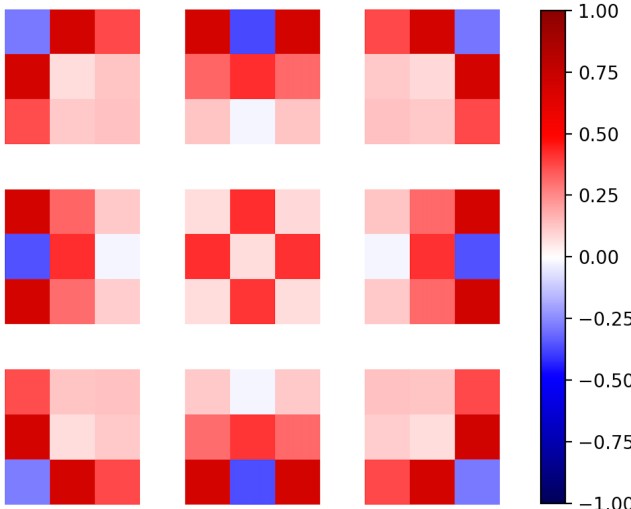

Figure 4: Visualization of the learned weights for pairwise potential defined in Equation 3 arranged according to the position of each patch within the $3 \times 3$ grid. The position of each sub-figure within the whole figure, represents the position of each patch within the $3 \times 3$ grid. The color map of each sub-figure shows the pairwise weights of all the patches with respect to the patch that the sub-figure represents. Weights were extracted from ResNet-18 with random seed 0.

NCRF improves the CNN feature extractor by incorporating spatial correlations between neighboring patches during training. Figure 3 shows the patch classification accuracies of the baseline method and NCRF based on the ResNet-18 architecture throughout training with different random seeds for initialization. NCRF consistently achieves higher training accuracies than the baseline method across different random seeds for initialization. Table 1 shows the best patch classification accuracies of the baseline method and NCRF on the validation set across different random seeds. NCRF consistently improves the patch classification accuracies on both ResNet-18 and ResNet-34, compared to the baseline method. These results show the CNN feature extractor benefits from the end-to-end joint training with CRF, compared to the previous two-stage training framework proposed in [15, 28], where the CNN feature extractor is not aware of the patch spatial correlations.

The learned weights for pairwise potential defined in Equation 3 also show strong spatial pattern. Figure 4 shows the visualization of the learned weights arranged according to the position of each patch within the $3 \times 3$ grid. For example, the sub-figure in the center represents the center patch

Table 1: Patch classification accuracies on the validation set.

|  | baseline | NCRF |
| --- | --- | --- |
| ResNet-18[12] | $0.9242 \pm 0.0007$ | $0.9296 \pm 0.0013$ |
| ResNet-34[12] | $0.9251 \pm 0.0007$ | $0.9338 \pm 0.0014$ |

Table 2: Average FROC score on the test set.

|  | baseline | NCRF |
| --- | --- | --- |
| ResNet-18[12] | $0.7825 \pm 0.0102$ | $0.7934 \pm 0.0168$ |
| ResNet-34[12] | $0.7444 \pm 0.0121$ | $0.7704 \pm 0.0171$ |

within the grid, and its color map shows the pairwise weights of all the patches with respect to the center patch. We can see that for a specific patch within the grid, its closest neighboring patches have the largest pairwise weights, suggesting the label distribution of each patch is strongly correlated with their closest neighbors within the grid.

### 3.5 NCRF Improves Cancer Metastasis Detection

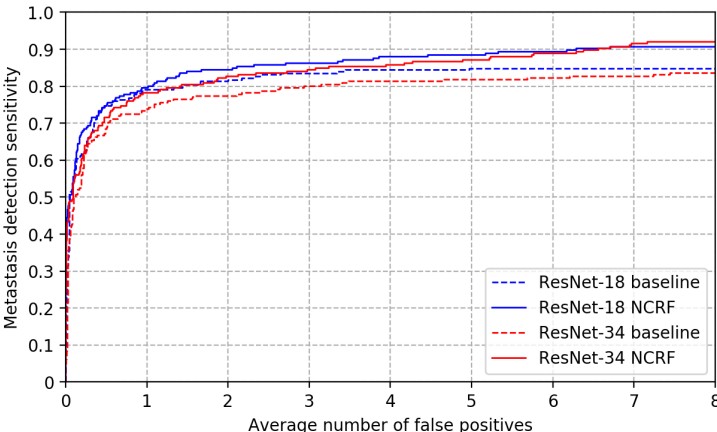

Figure 5: FROC curves of the baseline method and NCRF on the test set based on ResNet-18 and ResNet-34 with random seed 0.

We evaluated the performance of cancer metastasis detection of NCRF based on the average free-response receiver operating characteristic (FROC) score on the Camelyon16 test set. Given a list of predicted coordinates of cancer metastasis, the average FROC score is defined as the average detection sensitivity at 6 predefined false-positive rates per slide: $1/4$, $1/2$, $1$, $2$, $4$ and $8$. Higher average FROC score means better detection performance. We used the non maximum suppression algorithm [18], to obtain the coordinates of cancer metastasis based on a given probability map.

Figure 5 shows the curves of FROC scores of the baseline method and NCRF on the test set based on ResNet-18 and ResNet-34 with random seed 0. Table 2 shows the average FROC scores of the baseline method and NCRF on the test set based on ResNet-18 and ResNet-34 across different random seeds. NCRF also consistently improves the FROC scores on both architectures, compared to the baseline method. The best average FROC score from NCRF is $0.8096$ based on ResNet-18 with random seed 0, which outperforms the previous best average FROC score of $0.8074$ reported in [27].

## 4 Discussion

In this paper, we propose a neural conditional random field (NCRF) framework to detect cancer metastasis in Whole-slide Images (WSIs). NCRF is able to consider the spatial correlations between neighboring patches through the fully connected CRF component. Compared to previous methods, the CRF component is unified with the CNN feature extractor, and the whole model can be trained end-to-end with standard back-propagation algorithm. Because of this joint training framework, the CNN feature extractor also benefits from considering the spatial correlations while the CRF component introduces minor computational overhead. Compared to the baseline method without considering patch spatial correlations, NCRF obtains not only smoother probability maps but also better performances in cancer metastasis detection. NCRF is also a general technique that can be applied to other settings in pathology analysis, e.g. multiple-instance learning [14], when only whole slide level annotation is available. One future direction is using a grid of more than $3 \times 3$ patches as input, since it corresponds to a larger receptive field and may achieve better performance in cancer metastasis detection. Another interesting future direction is to compare NCRF with the baseline method that also uses patch coordinates as input, since patch localization is explicitly modeled in this case.

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
