# OpenReview forum: "Cancer Metastasis Detection With Neural Conditional Random Field"
_MIDL.amsterdam/2018/Conference — MIDL 2018 Oral_

### Review · AnonReviewer3 · 2018-05-06
**A CRF-based approach to go beyond single patch processing in WSI analysis with a fair evaluation**

**Rating:** 3
**Confidence:** 2

**Review:**

This work applies an end-to-end CRF-CNN for detecting metastasis in lymph nodes on high-dimensional whole-slide histology data for breast cancer. The CNN part is applied to individual image patches while the CRF work on the output of the CNN for a small grid of patches. The comprehensively validated results on an open data set supports the efficacy of the proposed method.

End-to-end training a CNN followed with a CRF to take into account the spatial correlation explicitly is not methodologically novel. However, applying this on the high-dimensional WSI seems well-motivated and, more importantly, showed improved performance due to the added CRF. The methods and experiments are well-organised and clearly-presented. The paper does not quite stress the small size of the grid of patches used for the CRF part. This is probably due to memory restrictions but it questions the need for large trainable regularisation. It would be interesting to better evaluate the impact of training the CRF as opposed to using CRF as a post-processing. This option would indeed limit the memory requirements.

The results are not reaching reported best numerical performance, this may invalidate the clinical relevance from the observed improvement due to the added CRF. Although likely given large validation/test data sizes, it helps to report the variance for the results from Table 1 and 2, in addition to appropriate statistical tests with p-values, to support the claimed significance. The numerical results, including the improvement has not been contexturised under the clinical application of interest.

A few minor comments I do not think need to be fully addressed as a condition to accept this work:
“Significant discordance on detection results among different pathologists has also been reported.” This is an interesting area pointed out by the authors, but I am disappointed not to see any further discussion or quantitative analysis on this point with the obtained results, such as detailed discrepancy (e.g. size, CI, clinical consequence) between computational methods and those between machine and human, in this very task.

Is there a quantitative measure to support that “smoother probability maps allow pathologists to focus more on potential tumor regions and less on isolated outliers” is beneficial?

VGG and AlexNet are fine, but more refined networks have been demonstrated in medical image segmentation tasks with better results, including using other loss functions other than CE.
It would be interesting to see the effect of varying grid sizes, especially the target is the detection and localisation instead of pure classification.

It would be interesting to see if even fewer iterations in mean-field approximation could achieve equivalent improvement, as random patch sampling may implicitly consider some level of spatial correlation.

Is excluding the background patches explicitly a better idea than weighting them down in loss?

It would be interesting to see if more explicit learning of localisation (predicting the coordinates) would encourage the network to learn the spatial correlation which was presumably omitted when not using CRF.


**Special Issue:**

No

---

> ### Comment · ~Yi_Li1 · 2018-05-23
> **reply to reviewer 3**
>
> Thanks very much for such detailed comments about our work. For the questions:
>
> 1. The paper does not quite stress the small size of the grid of patches used for the CRF part. This is probably due to memory restrictions but it questions the need for large trainable regularisation. It would be interesting to better evaluate the impact of training the CRF as opposed to using CRF as a post-processing. This option would indeed limit the memory requirements.
>
> We agree that grid size of 3x3 is relatively small. However, as pointed out by the reviewer, due to GPU limit, we can only feed in no more than 100 256x256 patches for each batch. If we further increase the grid size to 5x5, the batch size is then only 4, which introduced very large variance for batch normalization. CRF as post-processing has been studied in [25]. But they reported the inference time was slow on CPU if they used large embedding size and they have to use feature selection to down sample the embedding size to 5. Therefore, we did not compare CRF as post-processing since the embedding size is significantly different.
>
> 2. The results are not reaching reported best numerical performance, this may invalidate the clinical relevance from the observed improvement due to the added CRF.
>
> We agree that our reported average FROC at the time of submission was significantly worse than state-of-art result reported in [24]. Therefore, we have been actively working on improving the performances of both the baseline method and the proposed NCRF method. Fortunately, our proposed method has now achieved average FROC of 0.8096 which outperformed the average FROC of 0.8074 from [24] after camelyon16 resubmission. We are also working on open source our method for re-use and reproduce.
>
> 3. Although likely given large validation/test data sizes, it helps to report the variance for the results from Table 1 and 2, in addition to appropriate statistical tests with p-values, to support the claimed significance.
>
> We agree single number in Table 1 and 2 does not strongly support the advantages of our proposed NCRF method compared to the baseline. For Table 1, we will add the whole curves of training accuracies of our proposed method and the baseline method in the final version, showing that our method was doing better than the baseline throughout the training. We are also re-runing our training multiple times with different initial random seeds, and we will try our best to include the updated results before the camera ready deadline.
>
> 4. “Significant discordance on detection results among different pathologists has also been reported.” This is an interesting area pointed out by the authors, but I am disappointed not to see any further discussion or quantitative analysis on this point with the obtained results, such as detailed discrepancy (e.g. size, CI, clinical consequence) between computational methods and those between machine and human, in this very task.
>
> Thanks for pointing out this imprecise claim.  [7] reported the overall concordance rate of diagnostic interpretations of participating pathologists was 75.3% (95% CI, 73.4%–77.0%; 5194 of 6900 interpretations).  We will add this into our final version.
>
> 5. Is there a quantitative measure to support that “smoother probability maps allow pathologists to focus more on potential tumor regions and less on isolated outliers” is beneficial?
>
> Thanks for pointing out this imprecise claim. We don't have quantitative measure for supporting this and we will remove this sentence from the final version.
>
> 6. VGG and AlexNet are fine, but more refined networks have been demonstrated in medical image segmentation tasks with better results.
>
> We agree VGG and AlexNet are relatively outdated compared to later ResNet architecture. In our latest results, we did use ResNet18 and achieved a much better average FROC of 0.8096.
>
> 7. It would be interesting to see if even fewer iterations in mean-field approximation could achieve equivalent improvement, as random patch sampling may implicitly consider some level of spatial correlation.
>
> Both the baseline method and our proposed method use the same way to sample grid of patches for each mini batch, which did implicitly include spatial information as in the context of batch training. The only difference between the baseline method and our proposed the method is that our method further adds CRF smoothing during training. The baseline method can also be viewed as CRF with 0 mean-field iteration.
>
> 8. Is excluding the background patches explicitly a better idea than weighting them down in loss?
>
> During the grid-based patch sampling, background patches may occasionally be included especially at the boundary of tissues. Hard-negative mining further reinforce this effect.
>
> 9. It would be interesting to see if explicit learning of localisation would encourage the network to learn the spatial correlation.
>
> This is a very interesting point and we will investigate it in our next work.

---

### Review · AnonReviewer1 · 2018-05-07
**Simple and practical CRF-based CNN framework**

**Rating:** 3
**Confidence:** 2

**Review:**

Summary: Author presented a neural network framework integrated with dense CRF model. Different with related work[25] which used CRF as a post-processing, NCRF model put the CRF at the top of CNN architectures which makes it possible to jointly train the CNN and CRF models, and decreases the computational cost as well. The experimental results also showed the higher classification and smoother visual results comparing to baseline methods.

Comments: This is a interest contribution that connect CRF directly to the CNN model. Unlike related approaches such as CRF-RNN, NCRF methods integrate CRF with CNN in a more simply and explicit way. It makes this framework more flexible to apply to other neural network architectures.


Questions:

1. Hope to see a complete and successful comparison with previous work[24].

2. Please give a more detailed explanation of Eq. 3, the weight of smoothness kernel[15] become zero?
How about the weight of position-related term in appearance term？ In Eq. 3, the position-related term is fixed to 1?
The original pairwise equation in [15] is more reasonable.

3. It is better to show the FROC in figure with other reference approaches, not only the mean score.


**Special Issue:**

No

---

> ### Comment · ~Yi_Li1 · 2018-05-23
> **reply to reviewer 1**
>
> Thanks for the comments and questions. For the questions:
>
> 1. Hope to see a complete and successful comparison with previous work[24].
>
> We realize our reported average FROC at the time of submission was significantly worse than [24]. Therefore, we have been actively working on improving the performances of both the baseline method and the proposed NCRF method. Fortunately, our proposed method has now achieved average FROC of 0.8096 which outperformed the average FROC of 0.8074 from [24] after camelyon16 resubmission. We are also working on open source our method for re-use and reproduce.
>
> 2. Please give a more detailed explanation of Eq. 3, the weight of smoothness kernel[15] become zero?
> How about the weight of position-related term in appearance term？ In Eq. 3, the position-related term is fixed to 1?
> The original pairwise equation in [15] is more reasonable.
>
> Sorry for the confusing wording in the original paper. The weight of smoothness kernel in Eq.3 did not become 0. It was the weight for position-related term becoming 0 if we include them during training. Therefore, we ignore the position related term. But on the other hand, the learned weights for smoothness kernel did capture positional patterns and we will include this result in the final version. There is a significant difference between our method and [15], that the nodes in fully CRF in [15] are each pixel, while the nodes are each patches in our case. Therefore, we only include the term that measures the similarities between patches using their CNN embeddings.
>
> 3. It is better to show the FROC in figure with other reference approaches, not only the mean score.
>
> Thanks for pointing out this. We will include the FROC in figure in the final version.

---

### Review · AnonReviewer2 · 2018-05-09
**A very interesting paper**

**Rating:** 4
**Confidence:** 3

**Review:**

Overall:
The paper presents an idea of using neural conditional random fields, a combination of neural networks with a conditional random fields on top, to learn spatial dependencies among parts of a slide (patches). The idea is not new but its application to medical imaging is very smart. The obtained results are solid and convincing. The analysis of the results is clear and sound.

Strengths:
+ The paper is very well-written and it is very easy to follow.
+ All ideas are clearly explained.
+ All experiments are solid and the results are analyzed in-depth.
+ The proposed idea is not new (see a remark below), however, it makes a lot of sense in the proposed approach.
+ The learning algorithm is simple (mean-field), nevertheless, it is fast and works well in practice. It is especially important from the practical point of view.

Remarks:
* Major
- How many times were the experiments repeated? The differences between the reported results in Table 1 are small and, therefore, it would be beneficial to provide also a standard error.
- The idea of combining neural networks and conditional random fields is not new and could be traced back to, for instance, the following paper:
Do, T. and Artieres, T. (2010). Neural conditional random fields, AISTATS 2010.

I suggest to refer to this paper and discuss it briefly in the introduction.

* Minor
- Since the proposed idea is closely related to the multiple-instance learning, it would be interesting to discuss it briefly in the introduction or the conclusion.

**Special Issue:**

Yes

---

> ### Comment · (anonymous) · 2018-05-23
> **reply to reviewer 2**
>
> Thanks for the comments of strength and remarks of our work. For the questions:
>
> 1. - How many times were the experiments repeated? The differences between the reported results in Table 1 are small and, therefore, it would be beneficial to provide also a standard error.
> At the time when the paper was submitted, all the experiments were done once. We agree single number in Table 1 does not strongly support the advantages of our proposed NCRF method compared to the baseline. Therefore, we will add the whole curves of training accuracies of our proposed method and the baseline method in the final version, showing that our method was doing better than the baseline throughout the training. We are also re-runing our training multiple times with different initial random seeds, and we will try our best to include the updated results before the camper ready deadline.
>
> 2. - The idea of combining neural networks and conditional random fields is not new and could be traced back to, for instance, the following paper:
> Do, T. and Artieres, T. (2010). Neural conditional random fields, AISTATS 2010.
> Thanks for pointing out this paper, and we will add a brief discussion about this paper in our updated introduction.
>
> 3. Since the proposed idea is closely related to the multiple-instance learning, it would be interesting to discuss it briefly in the introduction or the conclusion.
> Thanks for pointing out this connection, and we will add a brief discussion in our updated conclusion.

---

> > ### Comment · ~Yi_Li1 · 2018-05-23
> > **deanonymous**
> >
> > Accidentally selected anonymous for signature. This is just to reveal the posted comment was by the original author Yi Li

---

### Decision · Program_Chairs · 2018-05-15
**Paper53 Acceptance Decision**

Oral